# Comparing the timeliness and adequacy of antenatal care uptake between women who married as child brides and adult brides in 20 sub-Saharan African countries

Sunday A. Adedini[1,2]*, Sunday Matthew Abatan[1], Adesoji Dunsin Ogunsakin[1], Christiana Alake Alex-Ojei[1], Blessing Iretioluwa Babalola[1], Sarafa Babatunde Shittu[1], Emmanuel Kolawole Odusina[1], Lorretta Favour C. Ntoimo[1]

1 Faculty of Social Sciences, Demography and Social Statistics Department, Federal University Oye-Ekiti, Oye-Ekiti, Nigeria, 2 Programme in Demography and Population Studies, University of the Witwatersrand, Schools of Public Health and Social Sciences, Johannesburg, South Africa

* sunday.adedini@fuoye.edu.ng

## Abstract

### Context

Considering the persistent poor maternal and child health outcomes in sub-Saharan Africa (SSA), this study undertook a comparative analysis of the timing and adequacy of antenatal care uptake between women (aged 20–24 years) who married before age 18 and those who married at age 18 or above.

### Method

Data came from Demographic and Health Surveys of 20 SSA countries. We performed binary logistic regression analysis on pooled data of women aged 20–24 (n = 33,630).

### Results

Overall, the percentage of child brides in selected countries was 57.1%, with the lowest prevalence found in Rwanda (19.1%) and the highest rate in Chad (80.9%). Central and West African countries had the highest prevalence of child marriage compared to other sub-regions. Bivariate results indicate that a lower proportion of child brides (50.0%) had 4+ ANC visits compared to the adult brides (60.9%) and a lower percentage of them (34.0%) initiated ANC visits early compared to the adult brides (37.5%). After controlling for country of residence and selected socio-economic and demographic characteristics, multivariable results established significantly lower odds of having an adequate/prescribed number of ANC visits among women who married before age 15 (OR: 0.63, CI: 0.57–0.67, p<0.001), and women who married at ages 15–17 (OR: 0.81, CI: 0.75–0.84, p<0.001) compared to those who married at age 18+. Similar results were established between age at first marriage and timing of first ANC visit. Other interesting results emerged that young women who

and Health Survey program repository (www.dhsprogram.com) and can be accessed following the protocol outlined in the Methods section.

**Funding:** The authors received no specific funding for this work.

**Competing interests:** The authors have declared that no competing interests exist.

married earlier than age 18 and those who married at age 18+ differ significantly by several socio-economic and demographic characteristics.

## Conclusion

Efforts to improve maternal and child health outcomes in SSA must give attention to address the underutilization and late start of antenatal care uptake among child brides.

## Introduction

Targets 3.1; 3.3; and 5.3 of the United Nations' Sustainable Development Goals (SDGs), respectively aim to reduce the global maternal mortality ratio to less than 70 per 100,000 live births; reduce under-5 mortality to at least 25 per 1,000 live births; and end the practice of child, early and forced marriage by 2030 [1]. While the world has made significant progress on these development indicators, pregnancy-related deaths continue to be a serious public health concern in sub-Saharan Africa (SSA). Pregnancy and the period surrounding it remain highly precarious for millions of women and children in SSA [2,3]. Despite several global efforts, maternal and child health outcomes in the region are still appallingly poor. The risk of SSA women dying from pregnancy-related complications is highest globally, with 533 maternal mortality ratio per 100,000 live births [1,3]. SSA is one of the two world's regions (the other being South Asia) where it is most dangerous to be a mother. The two regions account for 86% of global maternal deaths, with SSA alone contributing about 68% of the global statistics on maternal mortality. Worse still, the region has the highest lifetime risk of maternal death (1 in 38), as against 1 in 240 in South Asia and 1 in 5,400 in high-income countries [4,5].

Although the world recorded accelerated progress in reducing child mortality during the 2000–2019 period, with 1 in 27 children dying before age five in SSA in 2019 compared to 1 in 11 in 1990 (UNFPA, 2020); nevertheless, more than 5 million under-five children died globally in 2019 alone, and more than 50% of these deaths occurred in SSA. Several factors have been attributed to poor maternal and child health outcomes in SSA, including weak health system [6], environmental factors [7], malnutrition [8], socio-economic and bio-demographic characteristics [9–11] and clinical/medical related factors[12,13]. Additionally, one important risk factor of poor maternal and child health established in the literature is child marriage. A growing body of literature adduced that the social and health consequences of child marriage are enormous. These include maternal-related death during childbirth [14,15], obstetric fistulas [15,16], high risk of premature birth and neonatal and child death and increased risk for sexually transmitted diseases [17,18].

Considering that girls who married early are exposed to a high risk of gender-based violence and abuse due to unequal power relations [19,20], we hypothesised in this study that antenatal care use is lower among women who married before age 18 compared to those who married at adult age 18 or older. While studies have shown that low-cost interventions such as antenatal care (ANC) have been proven to be effective in promoting maternal and child health, ANC uptake in terms of its timeliness and adequacy has been below the World Health Organization's recommended thresholds in many SSA countries. For instance, scholars have established low quality antenatal care in Ethiopia and Nigeria as in many other SSA countries [20,21].

While young women require timely and adequate antenatal care use because of its usefulness for early detection of pregnancy complications, we posit in this study that young women

aged 20–24 who married as child brides (marriage before age 18) are likely to have a late start and inadequate uptake of ANC; however, evidence is sparse on this. Hence, in this study, we did a comparative analysis of timeliness and adequacy of antenatal care use between young women (aged 20–24 years) who married before18 years and those who married at age 18 or older, using datasets from 20 sub-Saharan African countries selected based on geographical difference and availability of recent data.

## Literature review and theoretical framework

Early marriage is more common among women of low socio-economic status [22,23]. It is a driving force for early pregnancy, as girls who marry before age 18 are under pressure to prove their fertility quickly [24,25]. It disrupts the normal trajectory of life for young girls by cutting short, in most cases, their schooling and economic opportunities, and consequently reduce their quality of life, and results in lower agency among young women [26]. Also, children of adolescent mothers have a higher risk of low birth weight and subsequent mortality [2,27]. Younger mothers themselves have higher maternal morbidity and mortality outcomes compared to older mothers [28].

Despite the risks involved with pregnancy at very young ages, the literature shows that girls and women in SSA are generally less likely to use antenatal care during their pregnancies [29,30]. Older age at marriage increased the likelihood of institutional delivery for women in Indonesia, while there was a higher likelihood of antenatal care use, institutional delivery and postnatal care use among women who had their first children at above age 20 [31].

A study in Senegal using structural equation modelling found that the age at first marriage and gender-role attitudes mediated the relationship between education and skilled birth attendant use [32]. Also, in Uttar Pradesh, India, women's age at marriage in addition to religion moderated the effects of literacy and wealth on facility delivery [33]. However, some studies showed that age at first marriage and childbearing had no association with antenatal care utilisation [34,35] while other studies showed increased antenatal care use and early booking among younger mothers [36,37]. This thus shows a lack of consensus in the literature regarding the relationship between age at marriage and antenatal care use.

This study is underpinned by the Andersen Behavioural Model of Healthcare Utilization (ABMHU). The model examines factors that influence healthcare utilization in a population, and classifies them into three categories–predisposing, enabling and need factors. Predisposing factors are those demographic and socioeconomic characteristics that encourage, in this case, antenatal care use for younger women, such as their level of education and other socio-economic characteristics. Enabling factors are those that provide the means for mothers to use maternal healthcare, such as their marital status and household income. Need factors are physiological which make healthcare access necessary and are categorized as perceived need, where the individual recognizes the necessity of accessing healthcare and evaluate need, where the need for medical care has been recognized by health professionals. In this case, the need for antenatal care is recognized and encouraged to ensure safe delivery and the survival of both mother and child.

This study utilized a conceptual framework based on the theoretical insights from ABMHU. We posit that the predisposing factors are demographic and socioeconomic characteristics (such as education, wealth index, occupation, media exposure, and place of residence) which may influence the relationship between antenatal care and age at first marriage. Previous studies have established that the factors that predispose women to use antenatal care during pregnancy include higher educational level and urban residence as well as parity [38,39]. Also, other studies found that women autonomy, marital status and gender-based violence or

gender relation serve as enabling factors for reproductive health issues such as maternal health care use and fertility outcomes [40–43]). Based on the theoretical insights drawn from ABMHU, we hypothesised in this study that underutilization and late start of ANC use are more likely among child brides compared to adult-brides.

## Materials and methods

### Data source

The study used the most recent Demographic and Health Survey (DHS) data of 20 countries selected across the four regional blocs of SSA. Surveys for the selected countries were conducted between 2013 and 2019. As shown in Table 1, the total sample was 33,630 with samples ranging from 227 in South Africa to 4072 in Nigeria. The DHS employs a comparable methodology to elicit demographic and health information from nationally representative samples across countries. A stratified two-stage cluster design sampling technique was employed in selecting representative samples. Enumeration areas (EAs) served as the primary sampling unit while a whole listing of households was done in selected EAs to derive representative samples in each country.

Analysis for this study focused on a group of young women (aged 20-24years) which is mutually exclusively divided into two strata: child brides (those who had their first marriage before age 18) and adult brides (those who had their first marriage at age 18+). We utilized the women's dataset and restricted analysis to young women aged 20–24 years who were ever

**Table 1. Weighted sampled population and percentage distribution of child brides and adultbrides across the selected SSA countries.**

| Sub-region/ Country | Survey year | Age at first marriage | | Sample, n (N = 33,630) |
|---|---|---|---|---|
| | | Child brides (<18 years) | Adult-brides (18 or older) | |
| **East Africa** | | | | |
| Ethiopia | 2016 | 67.8 | 32.2 | 1501 |
| Kenya | 2014 | 42.1 | 57.9 | 3098 |
| Rwanda | 2014–15 | 19.1 | 80.9 | 862 |
| Tanzania | 2015–16 | 47.6 | 52.4 | 1542 |
| Uganda | 2016 | 49.5 | 50.5 | 2583 |
| **Central and West Africa** | | | | |
| Benin | 2017–18 | 47.9 | 52.1 | 1,805 |
| Cameroon | 2018 | 57.9 | 42.1 | 1191 |
| Chad Republic | 2014–15 | 80.9 | 19.1 | 2437 |
| Cote D'Ivoire | 2013–14 | 59.8 | 40.2 | 2247 |
| Ghana | 2014 | 53.6 | 46.4 | 571 |
| Liberia | 2020 | 55.9 | 44.1 | 652 |
| Mali | 2018 | 67.8 | 32.2 | 1404 |
| Nigeria | 2018 | 69.7 | 30.3 | 4072 |
| Sierra Leone | 2019 | 58.6 | 41.4 | 1274 |
| **Southern Africa** | | | | |
| Lesotho | 2014 | 32.6 | 67.4 | 670 |
| Malawi | 2015–16 | 56.2 | 43.8 | 3815 |
| Mozambique | 2015 | 68.5 | 31.5 | 1028 |
| South Africa | 2016 | 20.1 | 79.9 | 227 |
| Zambia | 2018 | 50.1 | 49.9 | 1555 |
| Zimbabwe | 2015 | 49.6 | 50.4 | 1090 |
| **Total (selected SSA countries)** | | **56.5** | **43.5** | **33,630** |

married and pregnant at least once. Evidence from the literature supports the use of age-group 20–24 as a typical age category for studies on child marriage[40,44].

## Variable measurements

The dependent variable considered in this study was antenatal care (ANC) uptake measured in two ways–(i) timing of ANC start and (ii) adequacy of ANC uptake–the former was defined as early ANC initiation if respondents had the first visit during the first trimester, coded as '1', and '0' otherwise; while the latter was defined as adequate ANC use if respondents had 4 + ANC visits, coded as '1' and '0' otherwise. The ANC grouping of 'less than 4 visits' and '4 + visit' is based on the recommendation of the World Health Organization which stipulates the latter category as a minimum required number of visits to ensure optimal maternal health and newborn outcomes. The ANC use relates to the index child (which is the most recent birth). DHS questionnaire captured ANC use in terms of frequency and timing of visits, thus permitting the analysis conducted in this study.

The key explanatory variable in this study is child marriage, measured as marriage before age 18. Those who had their first marriage before age 18 were regarded as child brides while those whose age at first marriage was 18+ were considered as adult brides. We grouped the respondents into three categories: (i) very early marriage at <15 years, (ii) marriage at ages 15–17 years, and (iii) marriage at age 18+'. This was to permit examining variations in antenatal care use between respondents who had very early marriage (at ages less than 15 years) and those who married at ages 15–17, and 18+. Age at marriage was considered as the primary exposure in this study because, in many African societies, marriage is closely linked with first pregnancy and childbirth. Scholars have argued that the strong desire to get pregnant and have children soon after marriage is a significant driver of high fertility in many traditional societies [14,24,45]. We could not use variables such as age at first pregnancy and age at first birth because the former is not available in the DHS while the latter is only available for a few countries. Other independent variables considered in this study based on the reviewed literature and our theoretical/conceptual framework include current age (treated as a count variable), education (categorized as none, primary, and secondary/higher), religion (grouped as Christianity, Islam, and others), occupation (grouped as professional, sales/services, agriculture/others, currently not working), wealth index (categorized as poorest, poorer, middle, richer and richest), media exposure (no exposure, and had exposure), parity (grouped as 1 child, 2 children 3 or more children) and place of residence (urban and rural).

## Statistical analysis

Data were pooled from the selected 20 countries with a total sample size of 33,630. We analysed the data at three levels–univariate, bivariate, and multivariable analyses. Percentages, charts and frequencies were presented at the univariate level. We examined the relationship between the outcome measures and the key explanatory variables using the Chi-square test and cross-tabulation at the bivariate level. At the multivariable level, we employed binary logistic regression analysis to examine the statistically significant relationships between the outcome variable and the selected explanatory variables. Binary logistic regression was considered suitable for the analysis considering that the two outcome measures were binary taking the value of 1 if the event of interest happened and 0 otherwise; while the independent variables were categorical. Further, the stringent assumptions of linearity and normality of the dependent variable and the residuals could be relaxed for binary logistic regression, unlike the regression analysis. The binary logistic regression model is based on log transformation of

odds whose general equation is of the form:

$$\log\left(\frac{p}{1-p}\right) = \beta_0 + \beta_1 X_1 + \beta_2 X_2 + \cdots \beta_n X_n$$

Where; $\beta_0$ is the intercept

$\beta_i$ is the regression coefficient for the predictor variable $X_i$

$X_1$-$X_n$ are the predictor variables

$\left(\frac{p}{1-p}\right)$ is the odds of an event taking place. The ratio of odds of an event occurring in one group to the odds of the same event of interest occurring in the other group is called the odds ratio (OR). It indicates whether the odds of an event happening in one group are different from the odds of the same event taking place in another group, with an OR of 1 implying no difference in the event of interest between a given category of a predictor variable and the reference group.

We explored the predictors of each of the two outcome variables (timing of first ANC visit and adequacy of ANC use) by fitting four binary logistic models. Model 1 was the unadjusted model. Model 2 incorporated the key explanatory variable (age at first marriage) and selected demographic and socio-economic characteristics. Model 3 excluded education and added wealth index to the variables considered in Model 3 (to avoid multicollinearity between education and wealth index). Model 4 is the final model which incorporated all the selected variables into the analysis. The study's hypothesis was tested with a statistical significance set at $p<0.05$. At each level of analysis, datasets were weighted to adjust for the over or under-sampling of strata in sample selection. All analysis was done using Stata software (version 14.0). Also, descriptive analyses for the pooled data were weighted to adjust for the differences in sample size across countries. Multivariable analyses were performed with and without the use of sample size weight. Because results were similar with or without the use of sample size weight, only the unweighted multivariable analyses are presented. In the multivariable analysis, normative and largest groups were chosen as the reference categories. South Africa and Tanzania are missing in the regression models due to the fewness of observations.

## Results

### Descriptive analysis

Fig 1 shows the prevalence of child marriage and Table 1 presents the weighted sampled population and percentage distribution of child brides and adult brides across the selected SSA countries. On average, the percentage of child brides in selected SSA countries was 57.1%. As shown in Fig 1, the prevalence of child marriage varied considerably across countries and was lowest in Rwanda (19.1%) and SA (20.1%) and highest in the Chad Republic (80.9%). Central and West African countries had the highest prevalence of child marriage (47.9%-80.9%) while the percentage of adult brides was highest in the East African countries (32.2%-80.9%).

### Bivariate analysis

Table 2 presents the percentage distribution of age at first marriage by early ANC start; and adequate ANC use (4+ ANC visit). As shown in the table, age at first marriage was significantly associated with the timing of ANC visit (in 7 countries) and adequacy of ANC use in 13 of the countries studied ($p<0.05$). Overall (for all the selected countries), a slightly lower proportion of child brides initiated ANC visits early compared to the adult brides (34.0% vs. 37.5%). In the same vein, the percentage of child brides (50.0%) who had 4+ ANC visits was lower compared to the adult brides (60.9%). Results from the country-level analysis showed a similar pattern for almost all the selected countries. For instance, compared to the adult brides, the majority of

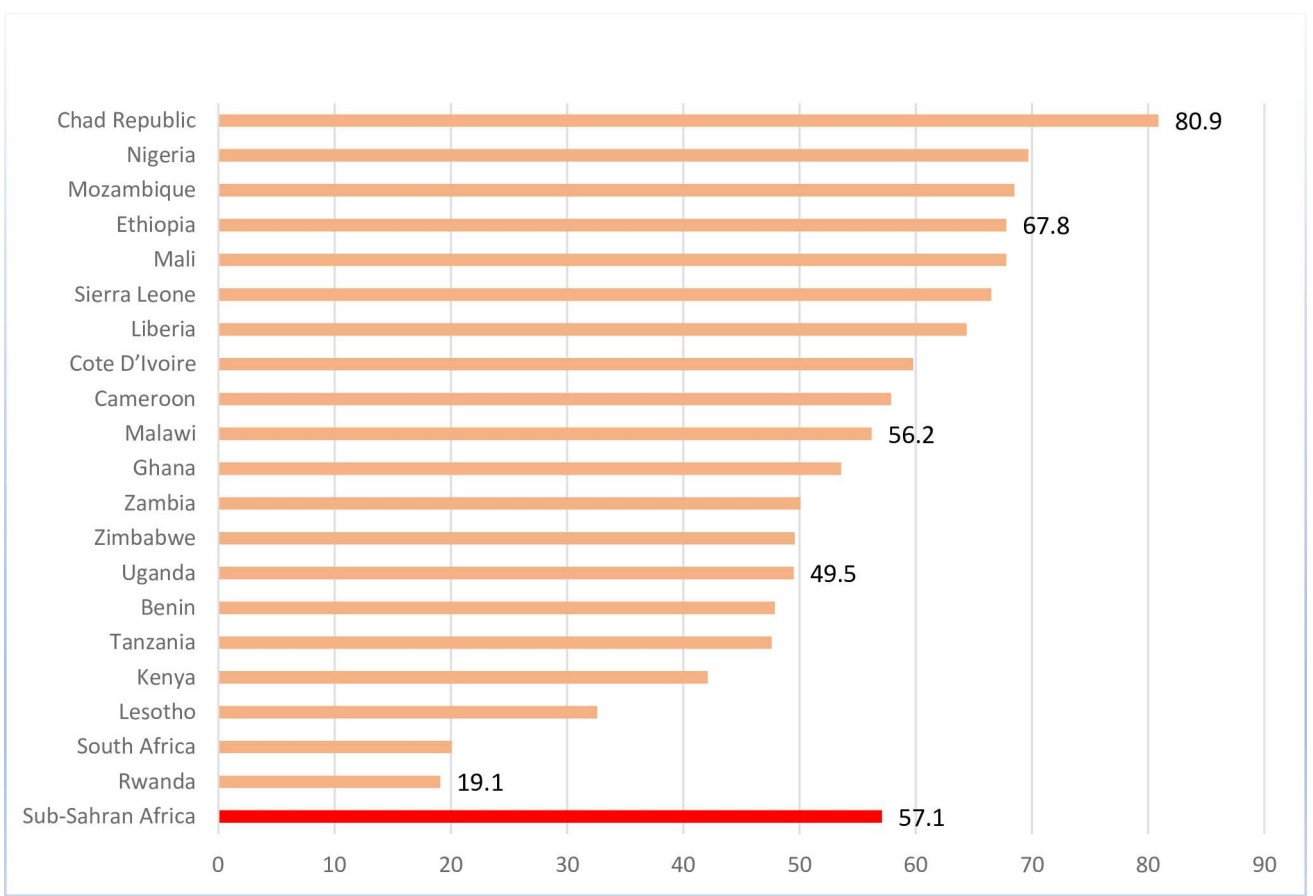

**Fig 1. Percentage of child-brides across selected SSA countries.**

child brides in almost all the countries had a lower proportion of women who initiated antenatal care early except in Rwanda, Benin, Ghana and Liberia. Mozambican dataset had no information on the timing of ANC visits.

The bivariate analysis/distribution of the study sample by age at first marriage and according to selected socio-economic characteristics is presented in Table 3. All the selected characteristics significantly varied by age at first marriage (p<0.05), thus showing that girls who married before age 15, those who married at age 15–17 and those who married at 18 or older age were significantly different by several socio-economic and demographic characteristics (including education, religion, occupation, household wealth status, media exposure and place of residence). The results indicate that the highest proportions of respondents aged 20-year-olds (49.1%), those with no formal education (45.7%), Muslim women (44.6%), women in agriculture/other petty occupations (41.4%), those from the poorest households (45.0%), women who had no media exposure (44.5%), multiparous women(48.5%), and rural women had their first marriage at ages 15–17.

## Age at first marriage as a predictor of adequate ANC uptake among married young women in sub-Saharan Africa

Table 4 presents results from the binary logistic regression analysis which examined the association between age at first marriage and adequacy of ANC use while adjusting for the selected

**Table 2. Percentage distribution of study participants aged 20–24 according to the timing of first ANC visit and number of ANC visit by selected SSA countries.**

| Sub-region/ Country | % with Early ANC start (at first trimester) | | | % with adequate ANC use (4+ ANC visit) | | |
|---|---|---|---|---|---|---|
| | Age at first marriage | | Significance ($X^2$) | Age at first marriage | | Significance ($X^2$) |
| | Child brides (<18) | Adult-brides (18+) | | Child brides (<18) | Adult-brides (18+) | |
| **East Africa** | | | | | | |
| Ethiopia | 20.8 | 19.1 | 0.63 | 26.1 | 35.8 | 15.34* |
| Kenya | 16.2 | 21.8 | 14.94** | 48.4 | 58.8 | 32.93*** |
| Rwanda | 62.6 | 65.9 | 0.63 | 44.8 | 52.0 | 2.67 |
| Tanzania | 22.2 | 27.1 | 4.86 | 45.3 | 53.9 | 11.26** |
| Uganda | 28.8 | 29.5 | 0.15 | 58.2 | 62.4 | 4.76 |
| **Central and West Africa** | | | | | | |
| Benin | 46.8 | 55.6 | 13.95** | 44.1 | 57.9 | 34.21*** |
| Cameroon | 32.4 | 47.2 | 26.76*** | 51.8 | 66.9 | 27.33*** |
| Chad Republic | 28.7 | 39.9 | 22.03** | 32.1 | 40.8 | 12.48** |
| Cote D'Ivoire | 16.5 | 18.3 | 1.23 | 45.3 | 48.6 | 2.31 |
| Ghana | 55.5 | 64.1 | 4.69 | 74.8 | 84.5 | 8.94* |
| Liberia | 66.9 | 63.4 | 0.95 | 76.8 | 81.0 | 1.89 |
| Mali | 36.1 | 42.6 | 5.32 | 40.1 | 45.7 | 3.87 |
| Nigeria | 13.0 | 25.1 | 90.03*** | 44.3 | 70.2 | 230.22*** |
| Sierra Leone | 41.2 | 38.1 | 1.23 | 72.8 | 73.8 | 0.15 |
| **Southern Africa** | | | | | | |
| Lesotho | 29.2 | 44.4 | 14.46*** | 61.37 | 74.9 | 13.1** |
| Malawi | 23.0 | 25.1 | 2.17 | 43.84 | 51.6 | 21.98*** |
| Mozambique | VNA | VNA | VNA | 46.3 | 54.8 | 6.13* |
| South Africa | 8.9 | 47.2 | 19.1*** | 53.0 | 73.8 | 6.34* |
| Zambia | 39.4 | 37.8 | 0.40 | 58.3 | 63.0 | 3.61 |
| Zimbabwe | 32.9 | 38.9 | 4.36 | 67.9 | 78.0 | 14.29*** |
| **Total (pooled data)** | 34.0 | 37.5 | 37.59*** | 50.0 | 60.9 | 386*** |

***$p<0.001$

**$p<0,01$

*$p<0.05$; VNA: Variable not available.

control variables. The results from the unadjusted model (Model 1a) showed significantly lower odds of using adequate ANC among women who had first marriage at ages 11 to 17 (OR = 0.51 to 0.80, p<0.05) compared to those who married at 18+.Unadjustedmodel examining the relationship between age at first marriage and adequate ANC use (Model 1b) also yielded similar findings. When the selected socio-economic and demographic characteristics were controlled for (without including country of residence), the association between age at first marriage and both indicators of ANC uptake became insignificant (analysis not shown). However, when the country of residence was incorporated into the model, the analysis showed a significant relationship between age at first marriage and the two indicators of ANC use. Model 2a (Table 5A) which adjusted for country of residence and selected socio-economic and demographic characteristics (excluding wealth index to guide against multicollinearity) indicates that young women who married before age 15 (OR: 0.62, CI: 0.58–0.68, p<0.01) and those who married at ages 15–17 (OR: 0.80, CI: 0.76–0.85, p<0.01) were significantly less likely to have adequate/prescribed number of ANC uptake compared to those who married at age 18 +.Model 3a that controlled for country, wealth index and other socio-economic and demographic variables (excluding education) and Model 4a (that incorporated all variables)

**Table 3. Percentage distribution of study participants aged 20–24 by age at first marriage and according to selected background characteristics (pooled data analysis).**

| Variables | % (Frequency) | Age at first marriage | | | Chi-square |
|---|---|---|---|---|---|
| | | <15 | 15–17 | 18+ | |
| **Current age** | | | | | |
| 20 | 20.9(7041) | 21.9 | 49.1 | 29.0 | |
| 21 | 16.1(5423) | 15.5 | 42.5 | 42.0 | 1033*** |
| 22 | 21.6(7246) | 16.4 | 40.7 | 42.8 | |
| 23 | 20.6(6917) | 13.6 | 36.8 | 49.7 | |
| 24 | 20.8(7002) | 12.9 | 33.4 | 53.7 | |
| **Education** | | | | | |
| None | 24.6(8278) | 30.1 | 45.7 | 24.2 | 3569*** |
| Primary | 41.4(13929) | 14.8 | 45.3 | 39.9 | |
| Secondary/Higher | 33.9(11422) | 7.6 | 30.6 | 61.8 | |
| **Religion[a]** | | | | | |
| Christianity | 66.19(22260) | 12.6 | 39.1 | 48.3 | 1309*** |
| Islam | 25.55(8592) | 26.5 | 44.6 | 28.9 | |
| Others | 2.9(984) | 18.6 | 42.1 | 39.3 | |
| **Occupation** | | | | | |
| Professional | 2.42(814) | 7.0 | 24.8 | 68.1 | 260*** |
| Sales/Service | 37.5(12,625) | 18.6 | 41.1 | 40.3 | |
| Agriculture/others | 19.6(6577) | 14.9 | 41.4 | 43.6 | |
| Currently not working | 40.5(13.613) | 16.5 | 40.1 | 43.4 | |
| **Wealth index** | | | | | |
| Poorest | 22.3(7494) | 20.6 | 45.0 | 34.4 | 975.31*** |
| Poorer | 22.99(7730) | 18.1 | 43.6 | 38.3 | |
| Middle | 20.0(6.796) | 16.7 | 41.6 | 41.6 | |
| Richer | 19.2(6448) | 12.3 | 37.5 | 50.2 | |
| Richest | 15.7(5262) | 10.5 | 31.4 | 58.1 | |
| **Media exposure** | | | | | |
| No exposure | 40.4(13.567) | 21.8 | 44.5 | 33.7 | 935.21*** |
| Had exposure | 40.5(20007) | 12.2 | 37 | 50.1 | |
| **Parity** | | | | | |
| 1 | 40.6(13567) | 4.9 | 29.2 | 65.9 | 5613.03*** |
| 2 | 36.6(12312) | 15.0 | 48.5 | 36.5 | |
| 3+ | 22.8(7672) | 37.8 | 47.5 | 14.8 | |
| **Place of residence** | | | | | |
| Urban | 28.8(9674) | 11.7 | 34.8 | 53.5 | 585.56*** |
| Rural | 71.2(23956) | 17.9 | 42.7 | 39.4 | |

***p<0.001.

produced similar findings Further analysis in Model 4a indicates some interesting findings. For instance, results revealed significantly lower odds of using adequate ANC in all countries compared to Nigeria (except Ghana, Lesotho, Liberia, Sierra Leone and Zimbabwe). Also, lower odds of utilizing adequate ANC were established among Muslim and rural women; while higher odds of having the prescribed number of ANC visits were established among women who had media exposure (OR = 1.22, CI: 1.16–1.29, p<0.001), women who had secondary or higher education (OR = 1.87, CI: 1.73–2.02, p<0.001), and among women from

**Table 4. Logistic regression model examining the influence of age at first marriage on adequacy and timing of first ANC among young married women in selected SSA countries.**

| Variables | Adequacy of ANC use | | Timing of ANC start | |
|---|---|---|---|---|
| | Model 1a | | Model 1b | |
| | Unadjusted OR | 95%CI | Unadjusted OR | 95%CI |
| Age at marriage | | | | |
| 10 | 0.51*** | (0.39–0.65) | 0.76 | (0.57–1.00) |
| 11 | 0.45*** | (0.36–0.56) | 0.70*** | (0.55–0.88) |
| 12 | 0.43*** | (0.37–0.50) | 0.63*** | (0.53–0.74) |
| 13 | 0.45*** | (0.40–0.51) | 0.51*** | (0.45–0.59) |
| 14 | 0.48*** | (0.44–0.52) | 0.58*** | (0.53–0.64) |
| 15 | 0.61*** | (0.57–0.65) | 0.67*** | (0.62–0.73) |
| 16 | 0.67*** | (0.62–0.71) | 0.77*** | (0.71–0.82) |
| 17 | 0.76*** | (0.71–0.81) | 0.80*** | (0.74–086) |
| 18+ (RC) | 1.00 | | 1.00 | |

OR = odds ratio, CI = confidence interval.

Model 1a examined relationship between age at first marriage and early ANC start.

Model 1b examined relationship between age at first marriage and adequate ANC use.

richest household (OR = 1.74, CI: 1.58–1.94, p<0.001) compared to those in the respective reference categories.

## Age at first marriage as a predictor of timing of ANC start among married young women in sub-Saharan Africa

Results from the binary logistic regression analysis which examined the association between age at first marriage and the timing of ANC initiation are presented in Table 5B. The results from Model 2bwhich controlled for the selected demographic and socio-economic characteristics indicate a significantly lower odds of having early ANC start among women who married before age 15 (OR: 0.76, CI: 0.69–0.83, p<0.001) and those who married at ages 15–17 (OR: 0.90, CI: 0.85–0.96, p<0.05) compared to those who married at age 18+. The results in the final model (Model 4b) further indicate a significantly higher odds of initiating ANC early in all countries (except Kenya and Cote D'Ivoire), among women who had media exposure (OR = 1.12, CI: 1.06–1.20, p<0.001), women with secondary or higher education (OR:1.54, CI: 1.41–1.67, p<0.001) and women in the richest households (OR = 1.72, CI: 1.55–1.92, p<0.001) compared to those in different reference groups. Also, Muslim women had significantly lower odds of initiating early ANC visit (OR = 0.77, CI: 0.71–0.83, p<0.05) compared to their Christian counterparts.

## Discussion

The main objective of this study was to examine how two antenatal care indicators (timing of first ANC visit and adequacy of ANC uptake) among young married women aged 20–24 compare between the child brides (women who married before age 18) and adult-brides (women who married at age 18 or above) in 20 sub-Saharan African countries. This study, which is one of the few attempts on the subject matter, is important considering that maternal deaths from pregnancy-related complications remain a serious public health concern in SSA. While antenatal care uptake holds great prospects for good maternal and child health outcomes [2], child brides in SSA face enormous poor health and developmental challenges [14,15], however,

**Table 5. A. Logistic regression model examining the influence of age at first marriage and selected characteristics on adequacy/number of ANC visit among young married women in selected SSA countries.** B. Logistic regression model examining the influence of age at first marriage and selected characteristics on timing of first ANC visit among young married women in selected SSA countries.

| Variables | Model 2a | | Model 3a | | Model 4a | |
|---|---|---|---|---|---|---|
| | Adjusted OR | 95%CI | Adjusted OR | 95%CI | Adjusted OR | 95%CI |
| **Age at First Marriage** | | | | | | |
| 18+ (RC) | | | | | | |
| 15-17 years | 0.80*** | (0.76-0.85) | 0.77*** | (0.73-0.82) | 0.81*** | (0.75-0.84) |
| <15 years | 0.62*** | (0.58-0.68) | 0.58*** | (0.54-0.64) | 0.63*** | (0.57-0.67) |
| **Country of Residence** | | | | | | |
| Nigeria (RC) | | | | | | |
| Benin | 0.92*** | (0.82-1.04) | 0.78*** | (0.69-0.88) | 0.87* | (0.78-0.99) |
| Cameroon | 0.94*** | (0.82-1.09) | 1.01 | (0.88-1.16) | 0.94 | (0.81-1.08) |
| Chad | 0.52*** | (0.47-0.59) | 0.42*** | (0.38-0.48) | 0.46*** | (0.42-0.53) |
| Cote D'ivoire | 0.59*** | (0.53-0.67) | 0.61*** | (0.55-0.69) | 0.58*** | (0.51-0.65) |
| Ethiopia | 0.53*** | (0.47-0.61) | 0.48*** | (0.43-0.56) | 0.48*** | (0.43-0.56) |
| Ghana | 3.00*** | (2.42-3.73) | 3.16*** | (2.55-3.95) | 3.19*** | (2.57-3.97) |
| Kenya | 0.71*** | (0.64-0.80) | 0.76*** | (0.68-0.85) | 0.73*** | (0.66-0.82) |
| Lesotho | 1.51*** | (1.25-1.83) | 1.74*** | (1.45-2.11) | 1.52*** | (1.26-1.84) |
| Liberia | 2.82*** | (2.32-3.43) | 2.93*** | (2.41-3.56) | 2.88*** | (2.37-3.51) |
| Malawi | 0.65*** | (0.59-0.73) | 0.67*** | (0.61-0.74) | 0.61*** | (0.55-0.68) |
| Mali | 0.58*** | (0.51-0.67) | 0.43*** | (0.38-0.50) | 0.54*** | (0.47-0.62) |
| Mozambique | 0.79** | (0.68-0.92) | 0.79*** | (0.68-0.92) | 0.74*** | (0.64-0.68) |
| Rwanda | 0.58*** | (0.50-0.69) | 0.58*** | (0.50-0.69) | 0.56*** | (0.48-0.67) |
| Sierra-Leone | 2.86*** | (2.48-3.31) | 2.87*** | (2.49-3.32) | 2.77*** | (2.40-3.21) |
| Uganda | 0.99 | (0.88-1.11) | 1.05 | (0.94-1.17) | 0.95 | (0.85-1.07) |
| Zambia | 1.05 | (0.92-1.21) | 1.19* | (1.04-1.36) | 1.06 | (0.94-1.22) |
| Zimbabwe | 1.49*** | (1.27-1.74) | 1.71*** | (1.46-2.00) | 1.49*** | (1.28-1.75) |
| **Age** | | | | | | |
| 20 (RC) | 1 | | | | | |
| 21 | 1.03 | (0.95-1.11) | 1.03 | (0.96-1.12) | 1.02 | (0.94-1.10) |
| 22 | 0.94 | (0.87-1.01) | 0.93 | (0.87-1.00) | 0.92* | (0.86-1.00) |
| 23 | 0.87** | (0.81-0.95) | 0.86*** | (0.80-0.93) | 0.85*** | (0.79-0.93) |
| 24 | 0.89** | (0.83-0.97) | 0.86*** | (0.80-0.94) | 0.86*** | (0.80-0.94) |
| **Religion** | | | | | | |
| Christianity (RC) | | | | | | |
| Islam | 0.83*** | (0.78-0.89) | 0.70*** | (0.66-0.75) | 0.82*** | (0.77-0.88) |
| Others | 0.81** | (0.71-0.93) | 0.77*** | (067-0.88) | 0.82*** | (0.71-0.94) |
| **Media Exposure** | | | | | | |
| Not Exposed (RC) | | | | | | |
| Exposed | 1.32*** | (1.25-1.39) | 1.29*** | (1.23-137) | 1.22*** | (1.16-1.29) |
| **Children Ever born** | | | | | | |
| 1 (RC) | | | | | | |
| 2 | 1.06* | (1.00-1.12) | 1.07* | (1.01-1.13) | 1.07* | (1.02-1.14) |
| 3+ | 1.11** | (1.04-1.20) | 1.12** | (1.05-1.22) | 1.14*** | (1.07-1.24) |
| **Residence** | | | | | | |
| Urban (RC) | | | | | | |
| Rural | 0.76*** | (0.72-0.80) | 0.87*** | (0.82-0.93) | 0.89** | (0.84-0.96) |
| **Occupation** | | | | | | |
| Not Working | | | | | | |

(*Continued*)

**Table 5.** (Continued)

| | | | | | | |
|---|---|---|---|---|---|---|
| Professional | 1.11 | (0.95-1.30) | 1.13 | (0.96-1.32) | 1.07 | (0.91-1.26) |
| Sales/Services | 1.15*** | (1.08-1.24) | 1.18*** | (1.10-1.27) | 1.14*** | (1.07-1.23) |
| Agriculture/Others | 1.02 | (0.97-1.09) | 1.05 | (099-1.12) | 1.05 | (1.00-1.12) |
| **Education** | | | | | | |
| None | | | | | | |
| Primary | 1.73*** | (1.61-1.85) | | | 1.67*** | (1.56-1.79) |
| Secondary+ | 2.05*** | (1.91-2.22) | | | 1.87*** | 1.73-2.02 |
| **Wealth Status** | | | | | | |
| poorest | | | | | | |
| poorer | | | 1.26*** | (1.18-1.35) | 1.18*** | (1.11-1.27) |
| middle | | | 1.36*** | (1.27-1.47) | 1.24*** | 1.15-1.34 |
| richer | | | 1.54*** | (1.42-1.67) | 1.36*** | 1.26-1.49 |
| richest | | | 2.02*** | (1.83-2.23) | 1.74*** | 1.58-1.94 |

*$p<0.05$; OR = odd ratio; CI = confidence interval.,—Cells are empty where models were not applicable.

Model 2a examined the relationship between age at first marriage and adequacy of ANC use while controlling for selected explanatory variables (excluding wealth status).

Model 3a achieves similar purpose but excludes education variable.

Model 4a examined same relationship while controlling for all the selected explanatory variables.

| Variables | Model 2b | | Model 3b | | Model 4b | |
|---|---|---|---|---|---|---|
| | Adjusted OR | 95%CI | Adjusted OR | 95%CI | Adjusted OR | 95%CI |
| **Age at First Marriage** | | | | | | |
| 18+ (RC) | | | | | | |
| 15-17 | 0.90* | (0.85 – 0.96) | 0.89*** | (0.84 – 0.95) | 0.90** | (0.85 – 0.96) |
| <15 | 0.76*** | (0.69 – 0.83) | 0.72*** | (0.66 – 0.79) | 0.76*** | (0.69 0.83) |
| **Country of Residence** | | | | | | |
| Nigeria (RC) | | | | | | |
| Benin | 4.92*** | (4.32 -5.61) | 4.27*** | (3.75 – 4.86) | 4.68*** | (4.11 – 5.33) |
| Cameroon | 2.43*** | (2.10 - 2.82) | 2.54*** | (2.19 – 2.94) | 2.43*** | (2.09 - 2.82) |
| Chad | 2.26*** | (1.99 -2.56) | 1.85*** | (1.63 – 2.10) | 2.01*** | (1.76 - 2.28) |
| Cote D'Ivoire | 0.77*** | (0.66 -0.88) | 0.76*** | (0.66 – 0.88) | 0.73*** | (0.63 – 0.85) |
| Ethiopia | 1.81*** | (1.57 -2.09) | 1.65*** | (1.43 – 1.90) | 1.64*** | (1.41 - 1.90) |
| Ghana | 5.96*** | (4.94 – 7.18) | 6.25*** | (5.18 – 7.53) | 6.30*** | (5.23 – 7.61) |
| Kenya | 0.74*** | (0.65 -.85) | 0.78*** | (0.69 – 0.90) | 0.75*** | (0.65 – 0.86) |
| Lesotho | 2.23*** | (1.86 -2.68) | 2.42*** | (2.01 – 2.90) | 2.22*** | (1.85 – 2.67) |
| Liberia | 7.72*** | (6.44 – 9.25) | 7.95*** | (6.64 – 9.52) | 7.88*** | (6.57 – 9.44) |
| Malawi | 1.22** | (1.08 -1.38) | 1.22*** | (1.08 – 1.37) | 1.12*** | (0.99 - 1.27) |
| Mali | 2.66*** | (2.29 -3.09) | 2.07*** | (1.79 -2.39) | 2.45*** | (2.11 – 2.85) |
| Mozambique | VNA | | | | | |
| Rwanda | 6.04*** | (5.09 – 7.22) | 6.11*** | (5.12 – 7.28) | 5.80*** | (4.85 – 6.93) |
| South Africa | – | | | | | |
| Sierra-Leone | 3.70*** | (3.21 – 4.26) | 3.68*** | (3.19 – 4.24) | 3.59*** | (3.11 – 4.14) |
| Tanzania | – | | | | | |
| Uganda | 1.43*** | (1.25 -1.62) | 1.46*** | (1.29 – 1.66) | 1.35*** | (1.18 – 1.53) |
| Zambia | 2.21*** | (1.92 -2.55) | 2.40*** | (2.08 – 2.76) | 2.22*** | (1.92 – 2.56) |
| Zimbabwe | 1.65*** | (1.41 -1.93) | 1.76*** | (1.50 – 2.06) | 1.63*** | (1.39 -1.91) |
| **Age** | | | | | | |
| 20 (RC) | | | | | | |

(*Continued*)

**Table 5.** (Continued)

| | | | | | | |
|---|---|---|---|---|---|---|
| 21 | 1.02 | (0.94 -1.11) | 1.03 | (0.94 – 1.12) | 1.02 | (0.93 - 1.11) |
| 22 | 0.96 | (0.89 – 1.04) | 0.95 | (0.88 – 1.03) | 0.95 | (0.87 - 1.03) |
| 23 | 0.93 | (0.85 - 1.01) | 0.91* | (0.83 – 0.98) | 0.90* | (0.83 – 0.98) |
| 24 | 0.94 | (0.86 – 1.03) | 0.91* | (0.83 - 0.99) | 0.91* | 0.83 – 0.99) |
| **Religion** | | | | | | |
| Christianity (RC) | | | | | | |
| Islam | 0.78*** | (0.72 -0.84) | 0.68*** | (0.64 – 0.74) | 0.77*** | (0.71 - 0.83) |
| Others | 0.76** | (0.64 – 0.89) | 0.73*** | (0.62 – 0.85) | 0.76** | (0.65 – 0.90) |
| **Media Exposure** | | | | | | |
| Not Exposed (RC) | | | | | | |
| Exposed | 1.21*** | (1.14 – 1.28) | 1.17*** | (1.10 – 1.24) | 1.12*** | (1.06 – 1.20) |
| **Children Ever born** | | | | | | |
| 1 (RC) | | | | | | |
| 2 | 1.01 | (0.94 -1.07) | 1.02 | (0.96 – 1.09) | 1.02 | (0.96 -1.09) |
| 3+ | 1.02 | (0.94 -1.11) | 1.06 | (0.97 – 1.14) | 1.06 | (0.97 – 1.15) |
| **Residence** | | | | | | |
| Urban (RC) | | | | | | |
| Rural | 0.85*** | (0.80 – 0.91 | 1.01 | (0.94 – 1.08) | 1.03 | (0.95 – 1.10) |
| **Occupation** | | | | | | |
| Not Working (RC) | | | | | | |
| Professional | 1.23* | (1.04 – 1.45) | 1.21* | (1.03 – 1.43) | 1.19* | (1.01 – 1.41) |
| Sales/Services | 1.10* | (1.01 – 1.17) | 1.11** | (1.03 – 1.19) | 1.08* | (1.01 - 1.17) |
| Agriculture/Others | 0.99 | (0.93 – 1.06) | 1.03 | (0.96 – 1.10) | 1.03 | (0.96 – 1.10) |
| **Education** | | | | | | |
| None (RC) | | | | | | |
| Primary | 1.58*** | (1.46 – 1.71) | | | 1.54*** | (1.42 – 1.67) |
| Secondary+ | 1.67*** | (1.53 – 1.81) | | | 1.54*** | (1.41 – 1.67) |
| **Wealth Status** | | | | | | |
| Poorest (RC) | | | | | | |
| poorer | | | 1.15*** | (1.06 -1.24) | 1.10* | (1.02 – 1.19) |
| middle | | | 1.24*** | (1.14 – 1.34) | 1.17*** | (1.07 - 1.27) |
| richer | | | 1.38*** | (1.26 -1.51) | 1.29*** | (1.18 – 1.41) |
| richest | | | 1.87*** | (1.69 – 2.08) | 1.72*** | (1.55 - 1.92) |

*p<0.05.; **p<0.01; ***p<0.001, OR = odd ratio; CI = confidence interval; VNA:—variable not available.

Cells are empty where models were not applicable.

Model 2b examined the relationship between age at first marriage and timing of ANC use while controlling for selected explanatory variables (excluding wealth status).

Model 3a achieves similar purpose but excludes education variable.

Model 4a examined same relationship while controlling for all the selected explanatory variables.

there is a dearth of research on the timeliness and adequacy of ANC use among child brides. Thus, we compared the timeliness and adequacy of antenatal care uptake between young women who married as child brides and those who married at adult age.

Our findings indicate that almost three-fifths of young women in selected countries married as child brides. Notwithstanding, there is sub-regional variation in the prevalence of child marriage in the selected countries, with Central and West Africa having the highest percentage of child marriage while East Africa generally had the highest proportion of women who had a first marriage as adults. This finding lends credence to prior research which established higher rates of child marriage in Central and West Africa compared to East Africa [3].

The study established some interesting variations, as many socio-economic and demographic characteristics differ significantly between respondents who married before age 18 and those who married at 18 or older age. These findings indicate that girls who marry at an early age are quite different in very many ways compared to those who marry at a later age. Compared with women with higher socioeconomic status, the prevalence of child marriage was higher among women with low education, women from poor households, women in non-professional occupations and the unemployed, multiparous women, and those in rural areas. It is a major finding from this study that many of the selected socio-economic and demographic characteristics that influence early marriage are also key drivers of poor and inadequate use of ANC in SSA.

Further, comparing the two indicators of ANC uptake, on the one hand, we observed a regional pattern as our bivariate results showed that in most of the selected countries in Central, West and Southern Africa, the percentage of women with early ANC start at the first trimester was significantly lower among child brides compared to the adult-brides. These results were re-echoed by our multivariable analysis. Particularly after adjusting for country of residence, we found that child brides were more likely to initiate ANC visits late compared to the adult-brides. We conjecture two plausible explanations for these findings. First, our data showed that child brides had poor access to media information, a situation that may affect their knowledge about reproductive health issues. Second, even if child brides have adequate knowledge regarding their reproductive health matters and know about the benefits of early ANC start, other findings from our analysis suggest that these women lack appropriate resources to receive adequate ANC from skilled healthcare personnel. As we posited, based on Andersen Behavioural Model of Healthcare Utilization, antenatal care use is largely influenced by women's socio-economic status. Findings from our analysis show that child brides have low educational attainment and poor socio-economic status, thus lending credence to findings of previous studies [22,23]. Due to their low socio-economic position in the household, child brides are subjected to control [46,47], and their partners/husbands and mothers-in-law wield considerable dominant power in the decision to seek care [9,16].

On the other hand, our bivariate analysis on the adequacy of ANC uptake reveals that overall and in each of the selected countries, a lower proportion of child brides had an adequate/prescribed number of ANC visits compared with adult-brides. These results were supported by the multivariable analysis, as our data indicate lower odds of having an adequate number of ANC visits among child brides compared to their counterparts who married at adult age. These findings reinforce the existing evidence regarding the negative implication of child marriage for maternal healthcare use. As established in extant literature [9,47,48], other factors that influence ANC use, including access to media/information, social isolation, limited access to household resources, poor access to quality care, poor socio-economic status and limited freedom of movement and autonomy in decision-making may play a significant role in these findings.

The concept of early marriage (which is marriage before 18 years in many contexts) is often viewed by scholars as a forced marital dyad [14,19,49]. Other studies have however reported different perspectives on the role of women's agency and autonomy and argued that early marriage is not just a determinant of poor socio-economic and health outcomes, but rather a viable option or response to it particularly among poor adolescent girls [50–52]. This is perhaps an important reason why similar socio-economic and demographic characteristics were established as drivers of both early marriage and poor ANC use.

The two indicators of antenatal care considered in this study appear to be interconnected, because women who initiated antenatal care visits late also largely have an inadequate number of antenatal care visits, perhaps because they had a late start at the second or third trimester.

Besides, the two indicators mainly have the same set of predictors such as education, media exposure, occupation, parity, religion, wealth index, and place of residence. Interventions towards ensuring early ANC start may therefore also lead to adequate ANC uptake among women.

The findings of this study have important policy implications. The observed underutilization and late start of antenatal care use among child brides make them an important subgroup of concern. With a very high proportion of child brides initiating antenatal care late in the second or third trimester as well as having an inadequate number of antenatal care visits to skilled healthcare providers, maternal and child mortality would remain high in SSA. As Izugbara and colleagues [3] noted, pregnancies and their outcomes would remain highly risky for child brides, except there are major interventions to address the problem. Since childbearing is closely linked to marriage in the African contexts [53], and child brides lack what it takes to negotiate safer and protected sex, millions of them would continue to get pregnant annually in a precarious condition. Moreover, considering that child marriage is mainly common among women of low socio-economic status [22,23], child brides largely lack economic resources and are unable to make independent decisions to seek healthcare. They are therefore at the mercy of their partners. We, therefore, recommend programmes and interventions that would lead to delayed marriage among girls, such as free and compulsory education. These would achieve two purposes–socio-economic empowerment of women and delay of first marriage among girls.

While interpreting the results of this study, it is important to consider some drawbacks, including the use of cross-sectional data which limits analysis only to exploring association. Also, the study was constrained to the analysis of variables available in the DHS, which hinders the investigation of culturally-laden drivers of early marriage and antenatal care non-use. Lastly, the datasets were not collected same year in different countries, albeit, this would not invalidate our findings, as evidence from the literature and our results demonstrate tardy progress in reducing child marriage in the selected countries during the period under study. Notwithstanding these limitations, the study has some strength and offers deeper insights on the implication of early girl-child marriage for a late start and inadequate ANC uptake. Besides, we analysed multi-country DHS data that utilized similar methodologies across the country, thus permitting comparability of study findings.

## Conclusion

This study concludes that efforts to reduce SSA's high rate of newborn and maternal mortality in line with the relevant SDG targets must give considerable attention to address the underutilization and late start of antenatal care uptake among child brides. Besides, considering the precarious conditions of child brides, including low education, poor employment prospects and generally low socio-economic status, interventions to improve the situations of girl-children must be a major focus and consideration of policymakers. Ultimately, there is the need to end girl-child marriage in SSA in line with target 5.3 which aims to end the practice of child, early and forced marriage by 2030.

## Acknowledgments

The authors thank the ICF International for the permission to utilize the Demographic and Health Survey data of the 20 selected countries. Reviewers' comments on the early draft of the manuscript are gratefully acknowledged. An earlier version of this article was presented at the 2021 International Population Conference of the International Union for the Scientific Study

of Population (IUSSP) held virtually from 5–10 December, 2021. Useful comments from the conference participants are gratefully acknowledged.

## Author Contributions

**Conceptualization:** Sunday A. Adedini.

**Formal analysis:** Sunday A. Adedini, Sunday Matthew Abatan, Blessing Iretioluwa Babalola.

**Methodology:** Sunday A. Adedini, Sunday Matthew Abatan, Blessing Iretioluwa Babalola.

**Supervision:** Sunday A. Adedini, Lorretta Favour C. Ntoimo.

**Writing – original draft:** Sunday A. Adedini, Sunday Matthew Abatan, Adesoji Dunsin Ogunsakin, Christiana Alake Alex-Ojei, Sarafa Babatunde Shittu, Emmanuel Kolawole Odusina, Lorretta Favour C. Ntoimo.

**Writing – review & editing:** Sunday A. Adedini, Emmanuel Kolawole Odusina, Lorretta Favour C. Ntoimo.

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
