## [Decision Letter · Decision Letter 0]

20 Jul 2021

PONE-D-21-15115

Comparing the timeliness and adequacy of antenatal care uptake between women who married as child-brides and adult-brides in 20 sub-Saharan African countries

PLOS ONE

Dear Dr. Adedini,

Thank you for submitting your manuscript to PLOS ONE. After careful consideration, we feel that it has merit but does not fully meet PLOS ONE’s publication criteria as it currently stands. Therefore, we invite you to submit a revised version of the manuscript that addresses the points raised during the review process. Please pay particular attention to the issues noted under "Additional Editors Comments" below.

We look forward to receiving your revised manuscript.

Kind regards,

David W Lawson

Academic Editor

PLOS ONE

Additional Editor Comments:

The reviewers note that the manuscript is well written, but highlight important concerns - that must be addressed carefully. Most importantly, reviewer 2 notes that your conclusions are NOT supported by the presented data. After controlling for a number of socio-economic and demographic characteristics (in particular woman’s education), the association between age at marriage and ANC use disappears (or at least loses significance / 95% CIs cross 1). This is reflected in Tables 4 and 5. However, in your discussion you indicate that the results are evidence that age at marriage is associated with ANC use (which it is only in bivariate analyses) in a way that implies causality. There is a sense of picking and choosing which results to discuss that fit an established narrative about the harms of early marriage, rather than examining the evidence objectively. Clearly, this is not acceptable. A more accurate conclusion may be that a number of socio-economic and demographic characteristics are associated both with age at marriage and ANC use, and as such may be driving both early marriage and poor ANC use. This is not addressed in the discussion or conclusion. According to Reviewer 2, you can not confidently conclude that ending child marriage will improve ANC use among young women based on the results as they stand.

A second important concern is that the category of 'child marriage' (and 'child bride') is adopted rather uncritically, with an assumption that all marriages under 18 years are inherently forced and that women have no autonomy in these unions, as an explanation for why poorer ANC use can be expected amongst younger women. I can recommend this paper by Schaffnit et al below (of which I am a coauthor) for a more grounded perspective on the drivers of early marriage and the role of women's agency. Your paper should consider why early marriage itself may not be best understood as a root cause of hardship, but rather a response to it - rather than have a forgone conclusion: Schaffnit SB, Wamoyi J, Urassa M, Dardoumpa M, & Lawson DW. (2020). When marriage is the best available option: perceptions of opportunity and risk in female adolescence in Tanzania. Global Public Health.

Both reviewers also make numerous suggestions about the analysis, which all must be considered fully. If you choose to revise and resubmit the manuscript I will examine these issues myself in close detail. For now, it is clear that author responses to a number of points are required before we can proceed. 

Thank you for submitting and I hope, despite some critical feedback, you find these points constructive and see a way forward with the paper!

Reviewers' comments:

Reviewer's Responses to Questions

**Comments to the Author**

1. Is the manuscript technically sound, and do the data support the conclusions?

Reviewer #1: Yes

Reviewer #2: No

2. Has the statistical analysis been performed appropriately and rigorously? 

Reviewer #1: No

Reviewer #2: Yes

3. Have the authors made all data underlying the findings in their manuscript fully available?

Reviewer #1: Yes

Reviewer #2: Yes

4. Is the manuscript presented in an intelligible fashion and written in standard English?

Reviewer #1: Yes

Reviewer #2: Yes

5. Review Comments to the Author

Reviewer #1: This is a useful piece of work. I suggest the following revisions:

1. The authors should add footnotes to show what each model in Tables 4 and 5 represent. The category first category for age in Table 3 (20) is different from that of Table 4 and 5 (<20). This should be corrected.

2. The authors pooled data from 20 SSA countries. It is not clear how weighting was applied. I do not mean the DHS sample weights but I mean the data weighting after pooling as you deal with multiple countries with wide variation in their population? (https://academic.oup.com/jn/article/128/10/1672/4723073?login=true)

3. The use of binary logistic regression does not cater for the hierarchical structure of the DHS datasets. I suggest the use of multilevel logistic regression

4. The authors did not control for country and survey year in their regression models. How do you deal with the heterogeneity in the 20 countries and the between and within country variations in the results?

5. I suggest presenting results for the final model of the regression outputs in the write-up of the results Again, using OR for both crude and adjusted results is confusing.

6. It is not clear why the authors chose to limit their study to DHS datasets up to 2018 when there are more recent datasets for countries like Liberia and Sierra Leone (All published in 2019).

7. I suggest the authors provide detailed information on the coding of the covariates in the methods.

8. It will be informative that the authors indicate what influenced the choice of the referent categories for the regression models. For instance, why was Christianity a referent category and Islam? I don’t want to assume that these were chosen because they were the first categories of the variables. The authors may refer to this link in deciding which category to choose as referent https://www.theanalysisfactor.com/strategies-dummy-coding/

9. Evidence from literature supports the use of age-group 20-24 as a typical age category for studies on child marriage (Godha et al., 2013; Yaya et al., 2019). The citation for this statement is different from the rest.

Reviewer #2: This was a well and clearly written paper that explores an important topic. The authors present results of original research and use appropriate data and analysis to address their research questions. My main concern is with the interpretation of results and discussion, which need to be revised before this manuscript can be recommended for publication.

Major comments

1. Introduction:

a. The authors mention that health risks of ‘child marriage’ are enormous, but surely these are the risks of early pregnancy, regardless of marriage? Some elaboration is needed here on why the authors chose to focus on age at marriage and not age at pregnancy or first birth as their primary exposure.

b. A clear conceptual framework is not developed. Which characteristics of early marriage do the authors think would be associated with poorer ANC use? They mention low SES, lower educational attainment, and pressures to prove fertility as factors that are associated with early marriage – but these are all factors that may be leading to early marriage?

2. Methods:

a. If the DHS allows, then why not measure age at marriage continuously to see variation between very early marriage (e.g., 13 years) and marriage closer to 18 years (e.g., 17 years). Marriage at 17 years is likely more similar to marriage at 18 years than marriage at 13 years, and as such there may be different effects on health seeking behaviour within the group under 18 years.

3. Results:

a. The authors state that age at marriage ‘was significantly associated with the timing of ANC visit and adequacy of ANC use in most countries studied’ – however, this is misleading. Table 2 shows that these findings were significant (p<0.05) for 7 out of 19 countries for early ANC start (i.e., 37% of the sample), and 11 out of 20 for 4+ ANC visits (55% of total sample), which I would not say is ‘most countries’. For initiating ANC visits early, the overall difference across all countries between those married before 18 years and those married at 18+ years is slim (34% vs 38%). There is a bigger difference for those who had 4+ ANC visits, but this too is limited to bivariate analyses.

b. I think the most interesting findings in this manuscript lie in Table 3. Here, the authors show that girls/women who marry before age 18 and those who marry at 18+ years differ significantly by a number of socio-economic and demographic characteristics: including education, occupation, household wealth index, media exposure, parity and urban/rural residence. This perhaps indicates, and is not reflected in the authors’ discussion at all, that girls who marry earlier are quite different in a number of ways to those who marry later and that these socio-economic and demographic characteristics may perhaps be driving BOTH early marriage as well as ANC use (and as such, that the relationship between age at marriage and ANC use is confounded by these socioeconomic and demographic characteristics). I suggest this is pulled out more in both the Results and Discussion sections.

c. My comment above is further supported by the authors’ main findings in the regression models. Firstly, education appears to drive the effect between age at marriage and adequate (4+ times) use of ANC. The association is also partially driven by other socio-economic and demographic factors included in the model – or in other words, all these factors are correlated with both age at marriage and adequate ANC use. Secondly, there is no association between age at marriage and timing of ANC use once socio-economic and demographic factors are controlled for. Again, the relationship between age at marriage and timing of ANC use appears to be driven by these socio-economic and demographic factors that are included in the models. These findings need to be brought out substantially in the discussion.

4. Discussion:

a. The discussion would be richer if it included details on the variation in age at marriage and ANC use across all SSA countries, instead of generalised statements like ‘almost three-fifths of young women in selected countries married as child brides’ as these can be misleading – which countries are they referring to? Again, it would be a lot more valuable to see variation in age at marriage under 18 years – how many of these women were marrying at age 17 (which is not so different from 18) and how many of them were marrying below that age, and especially how many (and in which countries) are marrying at very young ages (e.g., 13-14).

b. Equally, it would be valuable to examine age at first birth/pregnancy, and how that varies across regions and countries, as that is perhaps more directly related to ANC use than age at marriage.

c. A main strength of this paper is that the authors show which socio-economic and demographic characteristics of women are associated with ANC use, and which of these are also associated with early marriage. They in fact find no evidence to support their prediction that early marriage is associated with poor ANC use. However, in their discussion, they choose to present their bivariate findings instead of results from their final models which is misleading. Their recommendations are based on findings which they have themselves stated are not significant (p<0.05, Odds Ratios close to 1, and 95% confidence intervals crossing 1). I suggest this is revised.

d. Much of the literature cited in the discussion seems to suggest that poor educational attainment, low SES etc. are all associated with early marriage, but they do not consider that these factors may be what also lead to poor ANC use (e.g., issues of access and information may affect ANC use independent of age at marriage, and this is in fact shown in their regression models.)

e. Page 13, first paragraph, last sentence – more detail needed. Each of these factors should be discussed to show how they relate to both age at marriage and ANC use. I also think it is important to include a discourse here on the finding that after controlling for a number of socio-economic and demographic factors, age at marriage was in fact found to have no association with either measure of ANC use; and further that these socio-economic and demographic factors were strongly associated with ANC use and age at marriage. Why do the authors think these socio-economic and demographic factors are associated with both their exposure and outcome variables, and what are the possible pathways for these associations?

f. The authors do not give thought to why women might be marrying early and ignore contexts where women may be choosing to marry before age 18 (see Schaffnit, Urassa, Lawson, 2019; Stark, 2018; and others – citations given below). As such, there is some confusion between early marriage and forced marriage – whereas not all marriages before age 18 years are forced, or against the wishes of girls/women. By assuming that marriages before 18 years are forced or against women’s wishes, the authors take autonomy/agency away from these women, and simultaneously consider, for example, women marrying at age 18 (or slightly older) to have full agency, some of whom may not be very different to women aged 17 years and may also be entering marriages against their wishes. Again, important nuance would be added to this paper if the age at marriage variable distinguishes between very young girls and those who are closer to 18 years but still considered as ‘children’, as well as between those in the 18+ category (e.g., between an 18 year old and a 30 year old). I also recommend thinking a bit further about contextual differences in the meaning of ‘childhood’ as the definition of childhood can vary cross-culturally (see Hart, 2006; Rosen, 2007 - citations given below).

g. Further, a conceptual framework outlined in the introduction may help highlight which characteristics of marriage the authors expect will lead to poorer ANC use, distinguishing between, for example, forced marriage (and related loss of autonomy/agency), very early marriage (for example 13- or 14-year-olds), and age at first birth (and other aspects). As it stands, the authors are conflating these concepts under the umbrella of ‘child marriage’ / ‘marriage under 18-years’.

Minor comments

1. Terminology: I would suggest using the term ‘age at marriage’ throughout the paper to refer to the main exposure, instead of ‘child/adult marriage’. For example, in the Results section when discussing bivariate analyses, I think the authors mean ‘age at marriage’ was associated with ANC visits, not ‘child marriage’. Here, using the term ‘child marriage’ indicates that only marriage before 18 years is associated with ANC and not marriage at 18+ years – whereas the indicator that the authors are using is actually ‘age at marriage’. Using one term to describe their variable will add consistency to the manuscript.

2. Can this analysis be carried out using a linear variable for age at marriage – what is the effect of very early marriage (e.g., 13-15 year olds) versus 16-17 year olds versus 18-19 year olds? This would help differentiate the effects of very early marriage to marriage at ages closer to 18 years.

3. Abstract: The abstract needs to reflect the main findings - i.e., no association between age at marriage and ANC use after controlling for certain socio-economic and demographic variables.

4. Introduction:

a. What about younger unmarried woman - what is the rationale behind restricting analyses to married women, and why don’t the authors directly explore effects of early childbearing? An explanation can be added to the introduction.

b. The authors state that a high risk of gender-based violence among women married before age 18 would lead to lower ANC use – this may be true, but does not seem relevant to this paper. If it is relevant, the authors should clarify the links between age at marriage, gender-based violence and ANC use and elaborate on this relationship in detail, citing relevant literature.

3. Methods:

a. Clarity needed on the ANC measure for 4+ visits - what time frame does this refer to, the previous/most recent pregnancy?

4. Discussion:

a. Similar to minor comment 4b above: Stark statements are made about domestic violence and abuse with no clear explanation or rationale about how this relates to ANC use and early marriage – this topic is not explored in the authors' analyses, and these posited associations are not supported with relevant literature. I suggest this is revised either to fully explore how violence against women is associated with both ANC use and early marriage, with relevant citations, or the emphasis on violence against women is minimised as this is not directly relevant to this paper, i.e., the authors do not test the association between violence, ANC use and age at marriage.

b. The authors use causal terminology such as ‘child brides may initiate ANC visit late due to poor knowledge about their health issues’ without providing any supporting literature. Some of these statements come across as opinions and/or value judgements – citations are needed here, and even when literature is cited the different contexts of different studies (especially the age of women / age at marriage) need to be highlighted.

5. Papers of potential interest:

Dixon-Mueller, R. (2008). How young is “too young”? Comparative perspectives on adolescent sexual, marital, and reproductive transitions. In Studies in Family Planning (Vol. 39, pp. 247–262). Stud Fam Plann. https://doi.org/10.1111/j.1728-4465.2008.00173.

Hart, J. (2006). Saving children: what role for anthropology?, 1968(1), 131–134.

Rosen, D. M. (2007). Child Soldiers, International Humanitarian Law, and the Globalization of Childhood. American Anthropologist, 109(2), 296–306. https://doi.org/10.1525/aa.2007.109.2.296

Schaffnit, S. B., Hassan, A., Urassa, M., & Lawson, D. W. (2019, February). Parent–offspring conflict unlikely to explain ‘child marriage’ in northwestern Tanzania. Nature Human Behaviour. Nature Publishing Group. https://doi.org/10.1038/s41562-019-0535-4

Schaffnit, S. B., Urassa, M., & Lawson, D. W. (2019). “Child marriage” in context: exploring local attitudes towards early marriage in rural Tanzania. Sexual and Reproductive Health Matters, 27(1), 93–105. https://doi.org/10.1080/09688080.2019.1571304

Stark, L. (2018a). Early marriage and cultural constructions of adulthood in two slums in Dar es Salaam. Culture, Health and Sexuality, 20(8), 888–901. https://doi.org/10.1080/13691058.2017.1390162

Stark, L. (2018b). Poverty, Consent, and Choice in Early Marriage: Ethnographic Perspectives from Urban Tanzania. Marriage and Family Review, 54(6), 565–581. https://doi.org/10.1080/01494929.2017.1403998

6. PLOS authors have the option to publish the peer review history of their article (what does this mean?). If published, this will include your full peer review and any attached files.

Reviewer #1: **Yes: **Bright Opoku Ahinkorah

Reviewer #2: No

---

## [Author Response · Author response to Decision Letter 0]

8 Oct 2021

Response to reviewers’ comments

PONE-D-21-15115

Comparing the timeliness and adequacy of antenatal care uptake between women who married as child-brides and adult-brides in 20 sub-Saharan African countries

PLOS ONE

Comments

The reviewers note that the manuscript is well written, but highlight important concerns - that must be addressed carefully. Most importantly, reviewer 2 notes that your conclusions are NOT supported by the presented data. After controlling for a number of socio-economic and demographic characteristics (in particular woman’s education), the association between age at marriage and ANC use disappears (or at least loses significance / 95% CIs cross 1). This is reflected in Tables 4 and 5. However, in your discussion you indicate that the results are evidence that age at marriage is associated with ANC use (which it is only in bivariate analyses) in a way that implies causality. There is a sense of picking and choosing which results to discuss that fit an established narrative about the harms of early marriage, rather than examining the evidence objectively. Clearly, this is not acceptable. A more accurate conclusion may be that a number of socio-economic and demographic characteristics are associated both with age at marriage and ANC use, and as such may be driving both early marriage and poor ANC use. This is not addressed in the discussion or conclusion. According to Reviewer 2, you can not confidently conclude that ending child marriage will improve ANC use among young women based on the results as they stand.

Response

Thank you for the comments. We have now revised all sections of the manuscript. The multivariable analysis that was done previously showed an insignificant relationship between age at first marriage and antenatal care use. However, based on reviewers’ comments, we have undertaken a fresh analysis where we re-categorised age at first marriage to three groups in order to distinguish between respondents who married at a very early age (<15 years) and those who married at ages 15-17 and 18+. Our fresh analysis also controlled for the country of residence, and the results now show a significant relationship between age at first marriage and the two indicators of antenatal care uptake. The significant relationship between the outcome and exposure variables was as a result of re-categorizing age at first marriage and as well controlling for country of residence. The results demonstrate some between-country variations. We have discussed these findings in our discussion section.

Comments

A second important concern is that the category of 'child marriage' (and 'child bride') is adopted rather uncritically, with an assumption that all marriages under 18 years are inherently forced and that women have no autonomy in these unions, as an explanation for why poorer ANC use can be expected amongst younger women. I can recommend this paper by Schaffnit et al below (of which I am a coauthor) for a more grounded perspective on the drivers of early marriage and the role of women's agency. Your paper should consider why early marriage itself may not be best understood as a root cause of hardship, but rather a response to it - rather than have a forgone conclusion: Schaffnit SB, Wamoyi J, Urassa M, Dardoumpa M, & Lawson DW. (2020). When marriage is the best available option: perceptions of opportunity and risk in female adolescence in Tanzania. Global Public Health.

Response

We have read the suggested paper by Schaffnit et al (2020) as well as some other relevant publications and have considered the critical points raised as reflected in our revised texts presented below:

The concept of early marriage (which is marriage before 18 years in many contexts) is often viewed by scholars as a forced marital dyad (Adhikari, 2018; Mobolaji et al., 2020; Omariba & Boyle, 2007). Other studies have however reported different perspectives on the role of women’s agency and autonomy and argued that early marriage is not just a determinant of poor socio-economic and health outcomes, but rather a viable option or response to it particularly among poor adolescent girls (Al-Eisawi et al., 2021; Schaffnit et al., 2019; Schaffnit et al., 2020). This is perhaps an important reason why our analysis established similar socio-economic and demographic characteristics as drivers of both early marriage and poor ANC use. 

 Comments

We note that you have indicated that data from this study are available upon request. PLOS only allows data to be available upon request if there are legal or ethical restrictions on sharing data publicly. For more information on unacceptable data access restrictions, please see http://journals.plos.org/plosone/s/data-availability#loc-unacceptable-data-access-restrictions.

 Response

The data used for the manuscript are from Demographic and Health Survey. The datasets are available for public use. We have inserted the texts below into our cover letter:

Data for this manuscript are publicly available on the website of Demographic and Health Survey program (www.dhsprogram.org).

Reviewers' comments:

Reviewer's Responses to Questions

Comments to the Author

Reviewer #1: This is a useful piece of work. I suggest the following revisions:

1. The authors should add footnotes to show what each model in Tables 4 and 5 represent. The category first category for age in Table 3 (20) is different from that of Table 4 and 5 (<20). This should be corrected.

Response

The analysis covered women aged 20-24. We used the variable as a count measure. This is now consistently used in all the tables.

We have provided footnotes to indicate what each model represents.

Comments

2. The authors pooled data from 20 SSA countries. It is not clear how weighting was applied. I do not mean the DHS sample weights but I mean the data weighting after pooling as you deal with multiple countries with wide variation in their population? (https://academic.oup.com/jn/article/128/10/1672/4723073?login=true)

Response

We have included country sample weight in DHS cross countries analysis. According to Peng, et.al, (1998) in Maternal Nutritional Status Is Inversely Associated with Lactational Amenorrhea in Sub-Saharan Africa: Results from Demographic and Health Surveys II and III1–5, descriptive analyses for the pooled data were weighted to adjust for the differences in sample size across countries. The sample size weight was created from the equation: 1/(7.[nC/nT]), where nC is the sample size for each country and nT is the sample size for the pooled data.

Multivariable analyses were performed with and without the use of sample size weight. Because results were similar with or without the use of sample size weight, only the unweighted analyses are presented for the pooled analyses. We have included this information in the revised manuscript.

Comments

3. The use of binary logistic regression does not cater for the hierarchical structure of the DHS datasets. I suggest the use of multilevel logistic regression

Response

Thank you for this comment. Objective of this manuscript does not include exploring the influences of contextual characteristics on the outcome measures. This would be a good research question to interrogate in another manuscript.

Comments

4. The authors did not control for country and survey year in their regression models. How do you deal with the heterogeneity in the 20 countries and the between and within country variations in the results?

Response

The between and within country variations are expected in the results. The revised analysis has now considered country as a variable in our regression models. 

Comments

5. I suggest presenting results for the final model of the regression outputs in the write-up of the results Again, using OR for both crude and adjusted results is confusing.

Response

We have presented results for only the final model of the regression outputs in the write-up. UOR has now been used for the crude model and AOR is now used for the adjusted model. 

Comments

6. It is not clear why the authors chose to limit their study to DHS datasets up to 2018 when there are more recent datasets for countries like Liberia and Sierra Leone (All published in 2019).

Response

More recent datasets for selected countries have now been used in the revised analysis.

Latest Liberia and Sierra Leone already downloaded and added to the new analysis.

Comments

7. I suggest the authors provide detailed information on the coding of the covariates in the methods.

Response

We have provided detailed information in the method section on the coding of the selected variables.

Comments

8. It will be informative that the authors indicate what influenced the choice of the referent categories for the regression models. For instance, why was Christianity a referent category and Islam? I don’t want to assume that these were chosen because they were the first categories of the variables. The authors may refer to this link in deciding which category to choose as referent https://www.theanalysisfactor.com/strategies-dummy-coding/

Response

Thank you for this comment. In the multivariable analysis, normative and largest groups were chosen as the reference categories. We have included this information in the text

Comments

9. Evidence from literature supports the use of age-group 20-24 as a typical age category for studies on child marriage (Godha et al., 2013; Yaya et al., 2019). The citation for this statement is different from the rest.

Response

We have formatted the citations in accordingly 

Response Comments

Comments

Reviewer #2: This was a well and clearly written paper that explores an important topic. The authors present results of original research and use appropriate data and analysis to address their research questions. My main concern is with the interpretation of results and discussion, which need to be revised before this manuscript can be recommended for publication.

Major comments

1. Introduction:

a. The authors mention that health risks of ‘child marriage’ are enormous, but surely these are the risks of early pregnancy, regardless of marriage? Some elaboration is needed here on why the authors chose to focus on age at marriage and not age at pregnancy or first birth as their primary exposure.

Response 

Thank you. In response to this comment, we have provided some elaboration and explanation for focusing on early marriage/age at marriage. The revision is as shown below:

Age at marriage was considered as the primary exposure in this study because, in many African societies, marriage is closely linked with first pregnancy and childbirth. Scholars have argued that the strong desire to get pregnant and have children soon after marriage is a significant driver of high fertility in many traditional societies (Mobolaji et al., 2020; Spagnoletti et al., 2018). 

Comments

b. A clear conceptual framework is not developed. Which characteristics of early marriage do the authors think would be associated with poorer ANC use? They mention low SES, lower educational attainment, and pressures to prove fertility as factors that are associated with early marriage – but these are all factors that may be leading to early marriage?

Response

A relevant conceptual framework has been included in the revised manuscript. This is underpinned by the Andersen Behavioural Model of Healthcare Utilization. Based on this theory, we posit that antenatal care use is largely influenced by women’s low socio-economic status, lower educational attainment, etc. However, findings from our analysis show that early marriage is also associated with some of women characteristics such as low socio-economic status. We have discussed these findings in our discussion section.

Comments

2. Methods:

a. If the DHS allows, then why not measure age at marriage continuously to see variation between very early marriage (e.g., 13 years) and marriage closer to 18 years (e.g., 17 years). Marriage at 17 years is likely more similar to marriage at 18 years than marriage at 13 years, and as such there may be different effects on health seeking behaviour within the group under 18 years.

Response

In response to this comment, we did a fresh analysis using age at marriage as a count variable. We also have three categories for age at marriage: (i) very early marriage at <15 years’, (ii) ‘marriage at 15-17 years’, and (iii) ‘marriage at age 18+’. The new analysis shows variations in antenatal use and other characteristics between those who had very early marriage at <15 and those who married at 15-17, and 18+. These changes have been reflected in our method section, as well as in analysis and results.

Comments

3. Results:

a. The authors state that age at marriage ‘was significantly associated with the timing of ANC visit and adequacy of ANC use in most countries studied’ – however, this is misleading. Table 2 shows that these findings were significant (p<0.05) for 7 out of 19 countries for early ANC start (i.e., 37% of the sample), and 11 out of 20 for 4+ ANC visits (55% of total sample), which I would not say is ‘most countries’. For initiating ANC visits early, the overall difference across all countries between those married before 18 years and those married at 18+ years is slim (34% vs 38%). There is a bigger difference for those who had 4+ ANC visits, but this too is limited to bivariate analyses.

Response

Thank you for these comments. We have revised the manuscript in line with the suggestions. The results section now has the correct presentation of results. 

Comments

b. I think the most interesting findings in this manuscript lie in Table 3. Here, the authors show that girls/women who marry before age 18 and those who marry at 18+ years differ significantly by a number of socio-economic and demographic characteristics: including education, occupation, household wealth index, media exposure, parity and urban/rural residence. This perhaps indicates, and is not reflected in the authors’ discussion at all, that girls who marry earlier are quite different in a number of ways to those who marry later and that these socio-economic and demographic characteristics may perhaps be driving BOTH early marriage as well as ANC use (and as such, that the relationship between age at marriage and ANC use is confounded by these socioeconomic and demographic characteristics). I suggest this is pulled out more in both the Results and Discussion sections.

Response

Thank you for these suggestions. We have considered presenting these important results much more clearly in the results and discussion sections. All changes to the revised manuscript are tracked up. 

Comments

c. My comment above is further supported by the authors’ main findings in the regression models. Firstly, education appears to drive the effect between age at marriage and adequate (4+ times) use of ANC. The association is also partially driven by other socio-economic and demographic factors included in the model – or in other words, all these factors are correlated with both age at marriage and adequate ANC use. Secondly, there is no association between age at marriage and timing of ANC use once socio-economic and demographic factors are controlled for. Again, the relationship between age at marriage and timing of ANC use appears to be driven by these socio-economic and demographic factors that are included in the models. These findings need to be brought out substantially in the discussion.

Response

Based on the above suggestions, one of the major findings from this study is that many of the socio-economic and demographic characteristics that influence early marriage are also key drivers of low or inadequate uptake of ANC. These points have been clearly presented in the discussion section.

Comments

4. Discussion:

a. The discussion would be richer if it included details on the variation in age at marriage and ANC use across all SSA countries, instead of generalised statements like ‘almost three-fifths of young women in selected countries married as child brides’ as these can be misleading – which countries are they referring to? Again, it would be a lot more valuable to see variation in age at marriage under 18 years – how many of these women were marrying at age 17 (which is not so different from 18) and how many of them were marrying below that age, and especially how many (and in which countries) are marrying at very young ages (e.g., 13-14).

Response

Thank you for these suggestions. Considering them in the revised manuscript has made the discussion a lot better.

Comments

b. Equally, it would be valuable to examine age at first birth/pregnancy, and how that varies across regions and countries, as that is perhaps more directly related to ANC use than age at marriage.

Response

Age at first pregnancy is not available while age at first birth is only available for a few countries. Meanwhile, age at first marriage, as used in this study, is a good proxy for the two variables because marriage is closely linked to first pregnancy as women in many sub-Saharan African societies are under pressure and are expected to prove their fertility soon after marriage.

Comments

c. A main strength of this paper is that the authors show which socio-economic and demographic characteristics of women are associated with ANC use, and which of these are also associated with early marriage. They in fact find no evidence to support their prediction that early marriage is associated with poor ANC use. However, in their discussion, they choose to present their bivariate findings instead of results from their final models which is misleading. Their recommendations are based on findings which they have themselves stated are not significant (p<0.05, Odds Ratios close to 1, and 95% confidence intervals crossing 1). I suggest this is revised.

Response

Thank you for the comments. We have now revised all sections of the manuscript. The multivariable analysis that was done previously showed an insignificant relationship between age at first marriage and antenatal care use. However, based on reviewers’ suggestions, we have undertaken a fresh analysis where we re-categorised age at first marriage to three groups in order to distinguish between respondents who married at a very early age (<15 years) and those who married at ages 15-17 and 18+. Our fresh analysis also controlled for the country of residence, and the results now show a significant relationship between age at first marriage and the two indicators of antenatal care uptake. The significant relationship between the outcome and exposure variables was as a result of re-categorizing age at first marriage and as well controlling for country of residence. The results demonstrate some between-country variations. We have discussed these findings in our discussion section.

Comments

d. Much of the literature cited in the discussion seems to suggest that poor educational attainment, low SES etc. are all associated with early marriage, but they do not consider that these factors may be what also lead to poor ANC use (e.g., issues of access and information may affect ANC use independent of age at marriage, and this is in fact shown in their regression models.)

Response

The discussion has been revised to demonstrate that poor educational attainment and low socio-economic status are both drivers of early marriage and poor ANC use. We have also highlighted the roles of other factors that independently influence ANC use, such as the issues of information and access to access. These points have been clearly presented in the discussion.

Comments

e. Page 13, first paragraph, last sentence – more detail needed. Each of these factors should be discussed to show how they relate to both age at marriage and ANC use. I also think it is important to include a discourse here on the finding that after controlling for a number of socio-economic and demographic factors, age at marriage was in fact found to have no association with either measure of ANC use; and further that these socio-economic and demographic factors were strongly associated with ANC use and age at marriage. Why do the authors think these socio-economic and demographic factors are associated with both their exposure and outcome variables, and what are the possible pathways for these associations?

Response

The discussion has been revised to carefully reflect all the suggestions. We have presented a detailed discussion on how each of the selected factors relate to both age at marriage and ANC use.

Comments

f. The authors do not give thought to why women might be marrying early and ignore contexts where women may be choosing to marry before age 18 (see Schaffnit, Urassa, Lawson, 2019; Stark, 2018; and others – citations given below). As such, there is some confusion between early marriage and forced marriage – whereas not all marriages before age 18 years are forced, or against the wishes of girls/women. By assuming that marriages before 18 years are forced or against women’s wishes, the authors take autonomy/agency away from these women, and simultaneously consider, for example, women marrying at age 18 (or slightly older) to have full agency, some of whom may not be very different to women aged 17 years and may also be entering marriages against their wishes. Again, important nuance would be added to this paper if the age at marriage variable distinguishes between very young girls and those who are closer to 18 years but still considered as ‘children’, as well as between those in the 18+ category (e.g., between an 18 year old and a 30 year old). I also recommend thinking a bit further about contextual differences in the meaning of ‘childhood’ as the definition of childhood can vary cross-culturally (see Hart, 2006; Rosen, 2007 - citations given below).

Response

The revised manuscript now reflects the distinction between early marriages that are forced and the voluntary ones, thus giving consideration to women autonomy and agency. We have also included additional literature and presented a discussion on the contextual differences regarding the definition of childhood and reasons why young women marry early in different contexts. Also the revised analysis presents a clear distinction between very early marriage (before age 15) and those who married at 15-17.

Comments

g. Further, a conceptual framework outlined in the introduction may help highlight which characteristics of marriage the authors expect will lead to poorer ANC use, distinguishing between, for example, forced marriage (and related loss of autonomy/agency), very early marriage (for example 13- or 14-year-olds), and age at first birth (and other aspects). As it stands, the authors are conflating these concepts under the umbrella of ‘child marriage’ / ‘marriage under 18-years’.

Response

As suggested, a clear conceptual framework is now presented in the literature review section, clarifying the expected relationship between the selected characteristics and the outcome measures. 

Comments

Minor comments

1. Terminology: I would suggest using the term ‘age at marriage’ throughout the paper to refer to the main exposure, instead of ‘child/adult marriage’. For example, in the Results section when discussing bivariate analyses, I think the authors mean ‘age at marriage’ was associated with ANC visits, not ‘child marriage’. Here, using the term ‘child marriage’ indicates that only marriage before 18 years is associated with ANC and not marriage at 18+ years – whereas the indicator that the authors are using is actually ‘age at marriage’. Using one term to describe their variable will add consistency to the manuscript.

Response

Based on the suggestions, the revised manuscript now ensures consistency by using same terminology – ‘age at first marriage’ accordingly.

Comments

2. Can this analysis be carried out using a linear variable for age at marriage – what is the effect of very early marriage (e.g., 13-15 year olds) versus 16-17 year olds versus 18-19 year olds? This would help differentiate the effects of very early marriage to marriage at ages closer to 18 years.

Response

In response to this comment, we did a fresh analysis using age at marriage as a count variable. In the descriptive analysis, we also have three categories for age at marriage: (i) very early marriage at <15 years’, (ii) ‘marriage at 15-17 years’, and (iii) ‘marriage at age 18+’. The new analysis shows variations in antenatal use and other characteristics between those who had very early marriage at <15 and those who married at 15-17, and 18+. These changes have been reflected in our method section, as well as in the analysis, results and discussion.

Comments

3. Abstract: The abstract needs to reflect the main findings - i.e., no association between age at marriage and ANC use after controlling for certain socio-economic and demographic variables.

Response

The revised abstract now reflects results from our fresh analysis.

Comments

4. Introduction:

a. What about younger unmarried woman - what is the rationale behind restricting analyses to married women, and why don’t the authors directly explore effects of early childbearing? An explanation can be added to the introduction.

Response

The focus of the present paper is to explore whether child marriage is associated with ANC use. Our new analysis that considered additional variable has shown that there is a relationship between the two variables. Other interesting results also emerged and showed that young women and girls who married earlier than age 18 and those who married at age 18+ differ significantly by several socio-economic and demographic characteristics. We have presented these findings in the discussion section.

Comments

b. The authors state that a high risk of gender-based violence among women married before age 18 would lead to lower ANC use – this may be true, but does not seem relevant to this paper. If it is relevant, the authors should clarify the links between age at marriage, gender-based violence and ANC use and elaborate on this relationship in detail, citing relevant literature.

Response

We have presented a conceptual framework that clarifies the relationship between ANC use and selected characteristics. We did not examine the influence of gender-based violence on ANC use, therefore this variable has been removed from our discussion.

Comments

3. Methods:

a. Clarity needed on the ANC measure for 4+ visits - what time frame does this refer to, the previous/most recent pregnancy?

Response

The ANC grouping of ‘less than 4 visits’ and ‘4+ visit’ is based on the recommendation by the Word Health Organization which stipulates the latter category as a minimum required number of visits to ensure optimal positive maternal health and newborn outcomes. The ANC use relates to the index child (which is the most recent birth). These details have been included in the method section of the revised manuscript.

Comments

4. Discussion:

a. Similar to minor comment 4b above: Stark statements are made about domestic violence and abuse with no clear explanation or rationale about how this relates to ANC use and early marriage – this topic is not explored in the authors' analyses, and these posited associations are not supported with relevant literature. I suggest this is revised either to fully explore how violence against women is associated with both ANC use and early marriage, with relevant citations, or the emphasis on violence against women is minimised as this is not directly relevant to this paper, i.e., the authors do not test the association between violence, ANC use and age at marriage.

Response

More information and elaboration are now provided in the literature and conceptual framework. However, because we did not examine the influence of gender-based violence on ANC use, this variable has been removed from our discussion.

Comments

b. The authors use causal terminology such as ‘child brides may initiate ANC visit late due to poor knowledge about their health issues’ without providing any supporting literature. Some of these statements come across as opinions and/or value judgements – citations are needed here, and even when literature is cited the different contexts of different studies (especially the age of women / age at marriage) need to be highlighted.

Response

Relevant literature is now provided to support the statements made on the hypothesized relationship between ANC use and selected variables.

Comments

5. Papers of potential interest:

Dixon-Mueller, R. (2008). How young is “too young”? Comparative perspectives on adolescent sexual, marital, and reproductive transitions. In Studies in Family Planning (Vol. 39, pp. 247–262). Stud Fam Plann. https://doi.org/10.1111/j.1728-4465.2008.00173.

Hart, J. (2006). Saving children: what role for anthropology?, 1968(1), 131–134.

Rosen, D. M. (2007). Child Soldiers, International Humanitarian Law, and the Globalization of Childhood. American Anthropologist, 109(2), 296–306. https://doi.org/10.1525/aa.2007.109.2.296

Schaffnit, S. B., Hassan, A., Urassa, M., & Lawson, D. W. (2019, February). Parent–offspring conflict unlikely to explain ‘child marriage’ in northwestern Tanzania. Nature Human Behaviour. Nature Publishing Group. https://doi.org/10.1038/s41562-019-0535-4

Schaffnit, S. B., Urassa, M., & Lawson, D. W. (2019). “Child marriage” in context: exploring local attitudes towards early marriage in rural Tanzania. Sexual and Reproductive Health Matters, 27(1), 93–105. https://doi.org/10.1080/09688080.2019.1571304

Stark, L. (2018a). Early marriage and cultural constructions of adulthood in two slums in Dar es Salaam. Culture, Health and Sexuality, 20(8), 888–901. https://doi.org/10.1080/13691058.2017.1390162

Stark, L. (2018b). Poverty, Consent, and Choice in Early Marriage: Ethnographic Perspectives from Urban Tanzania. Marriage and Family Review, 54(6), 565–581. https://doi.org/10.1080/01494929.2017.1403998

Response

Thank you. The suggested papers have been reviewed and relevant ones have been cited accordingly.

Additional references included in the revised manuscript 

Adhikari, R. (2018). Child Marriage and Physical Violence: Results from a Nationally Representative Study in Nepal. Journal of Health Promotion, 6, 49-59. 

Al-Eisawi, Z., Jacoub, K., & Alsukker, A. (2021). A large-scale study exploring understanding of the national premarital screening program among Jordanians: Is an at-risk marriage a valid option for Jordanians? Public Understanding of Science, 30(3), 319-330. 

Mobolaji, J. W., Fatusi, A. O., & Adedini, S. A. (2020). Ethnicity, religious affiliation and girl-child marriage: a cross-sectional study of nationally representative sample of female adolescents in Nigeria. BMC Public Health, 20, 1-10. 

Omariba, D. W. R., & Boyle, M. H. (2007). Family Structure and Child Mortality in Sub-Saharan Africa: Cross-National Effects of Polygyny. Journal of Marriage and Family, 69, 528–543 

Schaffnit, S. B., Urassa, M., & Lawson, D. W. (2019). “Child marriage” in context: exploring local attitudes towards early marriage in rural Tanzania. Sexual and reproductive health matters, 27(1), 93-105. 

Schaffnit, S. B., Wamoyi, J., Urassa, M., Dardoumpa, M., & Lawson, D. W. (2020). When marriage is the best available option: Perceptions of opportunity and risk in female adolescence in Tanzania. Global public health, 1-14. 

Spagnoletti, B. R. M., Bennett, L. R., Kermode, M., & Wilopo, S. A. (2018). ‘I wanted to enjoy our marriage first… but I got pregnant right away’: a qualitative study of family planning understandings and decisions of women in urban Yogyakarta, Indonesia. BMC Pregnancy and Childbirth, 18(1), 353. https://doi.org/10.1186/s12884-018-1991-y

---

## [Decision Letter · Decision Letter 1]

7 Dec 2021

PONE-D-21-15115R1Comparing the timeliness and adequacy of antenatal care uptake between women who married as child-brides and adult-brides in 20 sub-Saharan African countriesPLOS ONE

Dear Dr. Adedini,

Thank you for submitting your manuscript to PLOS ONE. After careful consideration, we feel that it has merit but does not fully meet PLOS ONE’s publication criteria as it currently stands. Therefore, we invite you to submit a revised version of the manuscript that addresses the points raised during the review process.

ACADEMIC EDITOR:Dear authors on your scholarly work; you have brought an important study problem in the area of practice.However, the manuscript has some language usage flaws including punctuations, wordings, spelling and grammar errors. These problems are found throughout the manuscript. Moreover, there are some minor methodological limitations as the reviewer raised. Therefore, please make repeated proof-reading and thorough copyediting before resubmitting the manuscript. This would help increase the readability of the manuscript if published.

We look forward to receiving your revised manuscript.

Kind regards,

Wubet Alebachew Bayih, M.Sc.

Academic Editor

PLOS ONE

Reviewers' comments:

Reviewer's Responses to Questions

**Comments to the Author**

1. If the authors have adequately addressed your comments raised in a previous round of review and you feel that this manuscript is now acceptable for publication, you may indicate that here to bypass the “Comments to the Author” section, enter your conflict of interest statement in the “Confidential to Editor” section, and submit your "Accept" recommendation.

Reviewer #2: (No Response)

2. Is the manuscript technically sound, and do the data support the conclusions?

Reviewer #2: Yes

3. Has the statistical analysis been performed appropriately and rigorously? 

Reviewer #2: Yes

4. Have the authors made all data underlying the findings in their manuscript fully available?

Reviewer #2: Yes

5. Is the manuscript presented in an intelligible fashion and written in standard English?

Reviewer #2: Yes

6. Review Comments to the Author

Reviewer #2: The authors have done a good job in addressing the previous round of reviewer comments and the manuscript reads well now. Thank you for your detailed responses to my comments, much appreciated!

A few minor comments:

1. In the response to reviewer comments, you mentioned that data on age at first pregnancy is not available in the DHS and age at first birth was available for very few countries. I would add this point to your methods section too. You have added the point about why you used age at first marriage from a theoretical perspective but this will also clarify why you had to do this from a practical perspective.

2. Your conclusion states that from a policy perspective, ending early marriage is the solution to improving ANC use. Agreed, these two concepts are related, but perhaps also consider the idea that improving girls/women's socio-economic wellbeing, which - as acknowledged clearly in your discussion now - is a driver of both ANC use and age at marriage, should perhaps be the key focus of policy makers/programmes?

3. Table 2 - typo in heading: 'parentage' instead of 'percentage'

7. PLOS authors have the option to publish the peer review history of their article (what does this mean?). If published, this will include your full peer review and any attached files.

Reviewer #2: No

---

## [Author Response · Author response to Decision Letter 1]

13 Dec 2021

Response to Editor’s and reviewer’s comments

ACADEMIC EDITOR:

Comments

Dear authors on your scholarly work; you have brought an important study problem in the area of practice. However, the manuscript has some language usage flaws including punctuations, wordings, spelling and grammar errors. These problems are found throughout the manuscript. Moreover, there are some minor methodological limitations as the reviewer raised. Therefore, please make repeated proof-reading and thorough copyediting before resubmitting the manuscript. This would help increase the readability of the manuscript if published.

Response

Thank you for the comments. We have done a thorough editing of the manuscript and addressed the problems on punctuations, spelling and grammar errors. We have also addressed the minor methodological issues raised.

Reviewers' comments:

Reviewer #2: The authors have done a good job in addressing the previous round of reviewer comments and the manuscript reads well now. Thank you for your detailed responses to my comments, much appreciated!

Response

Thank you.

Comments

A few minor comments:

1. In the response to reviewer comments, you mentioned that data on age at first pregnancy is not available in the DHS and age at first birth was available for very few countries. I would add this point to your methods section too. You have added the point about why you used age at first marriage from a theoretical perspective but this will also clarify why you had to do this from a practical perspective.

Response

As advised, we have included the suggested information in the methods section of the manuscript as shown below:

We could not use variables such as age at first pregnancy and age at first birth because the former is not available in the DHS while the latter is only available for a few countries. 

Comments

2. Your conclusion states that from a policy perspective, ending early marriage is the solution to improving ANC use. Agreed, these two concepts are related, but perhaps also consider the idea that improving girls/women's socio-economic wellbeing, which - as acknowledged clearly in your discussion now - is a driver of both ANC use and age at marriage, should perhaps be the key focus of policy makers/programmes?

Response

Thank you for the suggestion. We have included the additional recommendation in the manuscript as shown below:

Besides, considering the precarious conditions of child brides, including low education, poor employment prospects and generally low socio-economic status, interventions to improve the situations of girl-children must be a major focus and consideration of policymakers.

Comments

3. Table 2 - typo in heading: 'parentage' instead of 'percentage'

Response

We have changed ‘parentage’ to ‘percentage’

---

## [Editor Report · Decision Letter 2]

3 Jan 2022

Comparing the timeliness and adequacy of antenatal care uptake between women who married as child-brides and adult-brides in 20 sub-Saharan African countries

PONE-D-21-15115R2

Dear Dr. Sunday A. Adedini,

We’re pleased to inform you that your manuscript has been judged scientifically suitable for publication and will be formally accepted for publication once it meets all outstanding technical requirements.

Kind regards,

Wubet Alebachew Bayih, M.Sc.

Academic Editor

PLOS ONE
---

## [Editor Report · Acceptance letter]

6 Jan 2022

PONE-D-21-15115R2 

Comparing the timeliness and adequacy of antenatal care uptake between women who married as child brides and adult brides in 20 sub-Saharan African countries 

Dear Dr. Adedini:

I'm pleased to inform you that your manuscript has been deemed suitable for publication in PLOS ONE. Congratulations! Your manuscript is now with our production department. 

Kind regards, 

on behalf of

Dr. Wubet Alebachew Bayih 

Academic Editor

PLOS ONE